# Functional Evaluation of Major Salivary Glands Using Viscosity PLUS and 2D Shear-Wave PLUS Elastography Techniques in Healthy Subjects—A Pilot Study

**DOI:** 10.3390/diagnostics12081963

**Published:** 2022-08-13

**Authors:** Delia Doris Muntean, Manuela Lavinia Lenghel, Diana-Raluca Petea-Balea, Anca Ileana Ciurea, Carolina Solomon, Sorin Marian Dudea

**Affiliations:** Department of Radiology, “Iuliu Hațieganu” University of Medicine and Pharmacy, 400012 Cluj-Napoca, Romania

**Keywords:** salivary glands, shear-wave, elastography, viscosity, healthy subjects

## Abstract

Biological soft tissues are characterized by viscoelastic properties. The propagation of shear waves within tissues is influenced by both elasticity, which is linked to the shear wave speed, and viscosity, which is linked to the shear wave dispersion. This study aimed to functionally assess the parotid glands (PG) and submandibular glands (SMG) in a group of 40 healthy subjects using the novel Viscosity PLUS (Vi.PLUS) and 2D Shear-Wave Elastography PLUS (2D-SWE.PLUS) techniques. The viscosity and stiffness of PG and SMG were measured before and after gustatory stimulation with a sialagogue agent (commercially available lemon juice) using the new SuperSonic MACH 30 ultrasound system equipped with a curvilinear C6-1X transducer. PG presented a mean basal viscosity and elasticity of 2.10 ± 0.19 Pa.s and 11.32 ± 1.91 kPa, respectively, which significantly increased poststimulation to 2.39 ± 0.17 Pa.s (*p* < 0.001) and 12.58 ± 1.92 kPa (*p* < 0.001), respectively. SMG did not present statistically increased values of viscosity and elasticity following stimulation (2.31 ± 015 Pa.s vs. 2.37 ± 0.18 Pa.s, *p* = 0.086, and 10.40 ± 1.64 kPa vs. 10.90 ± 1.98 kPa, *p* = 0.074, respectively). Vi.PLUS measurements presented a good positive correlation with 2D-SWE.PLUS values for PG and SMG, before and after stimulation. Gender and BMI were not confounding factors for these two parameters. Vi.PLUS represents an innovative non-invasive imaging technique that, together with 2D-SWE.PLUS proves to be useful in functionally assessing the major salivary glands in healthy subjects.

## 1. Introduction

Shear Wave Elastography (SWE) has been routinely employed to quantify the parenchymal stiffness of several organs and lesions, both qualitatively and quantitatively. Until recently, SWE imaging techniques have assumed the existence of a homogeneous, linear medium for the shear wave transmission process [1]. However, biological tissues present two mechanical properties: elasticity and viscosity, influencing the relationship between the delivered acoustic radiation force and tissue deformation, which proved to be nonlinear and time-dependent [2]. Elasticity is related to shear wave speed, which is linked to tissue fibrosis, while viscosity is related to shear wave dispersion, which proved to be influenced by inflammatory changes [3].

Only a few studies have been performed so far that have assessed viscosity as a new imaging parameter, mainly in chronic hepatopathies [4,5]. 

Reference viscosity values of major salivary glands (MSG) in healthy subjects have been proposed [6]. However, to the best of our knowledge, no functional studies using viscoelastography have been performed so far.

The objective of this study was to functionally assess the parotid glands (PG) and submandibular glands (SMG) by evaluating the parenchymal viscosity and stiffness variation and correlation before and after sialagogue stimulation in healthy subjects. Furthermore, this study also aims to provide reference values that could further guide future studies on inflammatory conditions affecting the MSG in adult populations. 

## 2. Materials and Methods

The current study was approved by the medical ethical committee of our university (DEP40/16.11.2021) and was performed following the World Medical Association Declaration of Helsinki (revised in 2000, Edinburgh). Informed consent was obtained from all subjects. Between December 2021 and January 2022, a prospective monocentric study including 40 healthy volunteers (16 men, 24 women, median age 30.4) was conducted. 

The inclusion criteria for all subjects were as follows: age older than 18 years; no medical history of head and neck surgery or radiotherapy; no medical history of chronic inflammatory connective tissue diseases (including Sjögren’s Syndrome), cystic fibrosis, sialolithiasis, or MSG tumors; no infectious diseases in the last three months before the study enrolment. Subjects were asked to fast for at least three hours before the examination. 

The real-time assessment of PG and SMG stiffness and viscosity was performed using the SuperSonic MACH^®^ 30 ultrasound system (SuperSonic Imagine, Aix-en-Provence, France). The ShearWave Elastography Plane-wave Ultrasound (2D-SWE.PLUS) and Viscosity Plane-wave Ultrasound (Vi.PLUS) modes were employed, available on the C6-1X curvilinear transducer. 

PG and SMG of each subject were examined in a standardized approach using anatomical landmarks by a single researcher with four years of experience in ultrasonography. All subjects were evaluated with their neck in hyperextension and the head oriented oppositely to the examiner. Longitudinal views of each gland were obtained, using a parallel plane to the posterior border of the vertical mandibular ramus for the PG and a parallel plane to the inferior border of the horizontal mandibular ramus for the SMG. Measurements were recorded for each gland in a basal condition (unstimulated). Then 5 mL of citric acid as commercially available undiluted lemon juice (unsweetened, with no additives or preservatives) was given orally using a 5 mL syringe, and 30 s later, the measurements were repeated. The lemon juice had to be maintained in the mouth for 10 s before being swallowed. 

2D-SWE.PLUS assesses tissue elasticity qualitatively, displaying a color-coded map superimposed on a B-mode image and quantitatively, allowing the local tissue stiffness measurement expressed in kPa over a broad range of values.

Vi.PLUS analyses the shear wave propagation speed at different frequencies providing information regarding shear wave dispersion inside tissues. The difference in shear wave velocity across frequencies is quantitatively quantified in Pa.s and displayed on a color-coded map.

Elasticity measurements are displayed as mean, median, minimum, maximum, and SD, while viscosity measurements are displayed as mean, median, and SD. Information regarding the depth and diameter of the Q-Box and the Stability Index values are also provided. (Figure 1).

### 2.1. Imaging Protocol

2D-SWE.PLUS mode can be combined with the Vi.PLUS mode; therefore, the acquisitions were made simultaneously for stiffness and viscosity measurements. 

All subjects were examined with the curvilinear C6-1X transducer. A generous amount of gel between the skin and the transducer was used. The imaging protocol was optimized by using the resolution setting and selecting the AutoTGC function after obtaining an optimal B-mode image. The 2D-SWE.PLUS and Vi.PLUS modes were then engaged, and the corresponding trapezoid color box was positioned in a homogenous, artifact-free area with no vessels or moving structures. All measurements were performed using a 5 mm circular Q-Box, positioned in the center of the color box at approximately 1.5 cm depth from the skin. The color box and the Q-Box are duplicated on both 2D-SWE.PLUS and Vi.PLUS images, allowing the stiffness and viscosity quantification at the same place and the same time.

The transducer was held with minimal pressure for at least 3 s to stabilize the image. Subjects were asked to hold their breath and avoid swallowing while taking the measurements to minimize motion-related artifacts. The stability index (SI) represents the quality parameter developed by SuperSonic Imagine and is derived from the temporospatial stability of stiffness and viscosity within the Q-Box. If the SI value is below 90%, the manufacturer suggests placing the Q-Box elsewhere in the region of interest or repeating the image acquisition. Thus, measurements were only recorded in the current study when the stability index (SI) was greater than 90%. The mean value of three valid measurements obtained in a homogenous area from three different frames was considered (quantified in kPa for elasticity and Pa.s for viscosity, respectively).

### 2.2. Statistical Analysis

The statistical analysis was performed using MedCalc Version 20 (MedCalc Software Corp., Brunswick, ME, USA) and IMB SPSS Statistics Version 23. The Kolmogorov-Smirnov test was used to assess the data distribution. Normally distributed data were presented as mean ± standard deviation and non-normally distributed as median and interquartile range. For the statistical comparisons, the paired-sample T-test was used, and Pearson’s or Spearman’s coefficient was computed for statistical correlations. The differences between gender groups were assessed using the Kruskal-Wallis one-way analysis of variance test. A 2-tailed *p*-value less than 0.05 was considered statistically significant. 

## 3. Results

This study included a total of 40 healthy subjects with a median age of 29 years old. The baseline characteristics are presented in Table 1. 

There were no differences between the values recorded for the left and right PG and SMG, before (*p* = 0.231, *p* = 0.120) or after stimulation (*p* = 0.445, *p* = 0.523). Therefore, for each subject, the averaged measurements of the right and left salivary glands were further included in the analysis.

In a basal state, PG of healthy subjects presented a significantly lower mean viscosity and a higher stiffness in comparison to SMG (2.10 ± 0.19 Pa.s versus 2.31 ± 0.15 Pa.s, *p* < 0.001, and 12.58 ± 1.92 kPa versus 10.40 ± 1.64 kPa, *p* = 0.023, respectively). 

Following gustatory stimulation, the mean Vi.PLUS values of PG increased significantly to 2.39 ± 0.17 Pa.s, *p* < 0.001. A similar trend was observed for the mean 2D-SWE.PLUS values of PG, which were also higher post-stimulation, increased up to 12.58 ± 1.92 kPa, *p* < 0.001.

The mean values of viscosity and elasticity for the SMG were also higher following stimulation (2.37 ± 0.18 Pa.s and 10.40 ± 1.64 kPa, respectively), but the difference was not statistically significant (*p* > 0.05) (Table 2, Figure 2).

The 2D-SWE.PLUS values of the PG presented a strong positive correlation with the Vi.PLUS values in basal conditions (r = 0.711, *p* < 0.001), and after stimulation, respectively (r = 0.676, *p* < 0.001). The 2D-SWE.PLUS values of the SMG also showed a moderate positive correlation with the Vi.PLUS values in basal conditions (r = 0.646, *p* < 0.001), and following stimulation, respectively (r = 0.625, *p* < 0.001) (Figure 3). 

There were no statistically significant differences in the mean viscosity and elasticity values of PG and SMG between gender groups before and after stimulation (*p* > 0.05) (Table 3). 

Also, no statistically significant correlation was found between the BMI and the viscosity and elasticity measurements of PG and SMG, in the basal state and following stimulation (*p* > 0.05), (Table 4). 

## 4. Discussion

Viscosity is a novel imaging parameter based on the shear wave dispersion process linked to inflammatory alterations within tissues [5]. SuperSonic Imagine allows the quantification of this new parameter using the Vi.PLUS mode, simultaneously engaged with the 2D-SWE.PLUS mode, which assesses tissue elasticity based on shear wave propagation speed. 

The normal baseline stiffness values for PG (11.32 ± 1.91 kPa) and SMG (10.40 ± 1.64 kPa) revealed in this study fall within the ranges reported in the literature. The normal SWE values for PG vary between 5.39–26 kPa and 8.15–22 kPa for SMG, respectively [7,8,9,10]. The recent EFSUMB guideline on performing elastography suggests there is variability in the shear wave speed values between different ultrasound vendors and used equipment, which is mainly related to system factors such as bandwidth and shear wave vibration mean frequency [11]. This might explain the broad spectrum of reported normal SWE values for PG and SMG. Consequently, vendor-specific cut-off values might be necessary. 

The normal baseline viscosity values in this study are similar to the ones obtained in one previous preliminary study conducted in our department [6] for the PG (2.10 ± 0.19 vs. 2.13 ± 0.23 Pa.s) and slightly lower for the SMG (2.31 ± 0.15 vs. 2.44 ± 0.35 Pa.s), differences possibly related to a narrower age range for the subjects included in the current study. 

Functional salivary glands assessment in healthy subjects using gustatory stimulation with lemon juice has also been previously performed in several studies [12,13,14,15,16]. Using high-temporal resolution echo-planar diffusion-weighted image (DWI), an increase in the apparent diffusion coefficient (ADC) value was proved seconds after oral administration of lemon juice [12]. An increased ADC rate of 0.25 ± 0.08 was also demonstrated for both PG and SMG [13]. ADC values of PG after stimulation proved to be higher using magnetic field strengths of 1,5T and 3T, respectively [14]. With Doppler ultrasonography, the submandibular blood flow (assessed by recording the facial artery’s maximum velocity) increased in response to stimulation [15]. 

So far, only one study has used an elastographic-based technique to assess MSG functionally. Tasdemir et al. [16] analyzed 30 patients with xerostomia using acoustic radiation force impulse imaging with virtual touch quantification. They demonstrated that the degree of stiffness decreased after the lemon juice stimulation in the PG and SMG. 

So far, no studies have been reported to address the functional changes of salivary glands using viscosity and 2D-SWE techniques, neither on healthy subjects nor on patients with MSG pathology. 

Our results show that the mean viscosity and stiffness increased significantly following gustatory stimulation compared to the PG’s basal state. At the same time, for the SMG, the values were only slightly higher, with no statistically significant difference. 

PG proved to be responsible for the majority production of stimulated saliva, while SMG and sublingual glands provide most of the saliva secretion in the basal state [17,18]. One study that functionally assessed the MSG of healthy subjects using dynamic MR sialography revealed that the PG ducts were detectable immediately after the citric acid stimulation, in contrast to SMG ducts which were slower detectable. The changing ratio was two folds higher in the PG ducts but not in the SMG ducts [19]. Thus, each gland’s physiological salivary secretion mechanism might explain the less significant change in SMG viscosity and stiffness values following stimulation, as opposed to PG.

Gender and BMI were not confounding factors of the Vi.PLUS and 2D-SWE.PLUS values in this study. This observation is in accordance with other studies performed on adult populations that assessed MSG using elastography [7,20,21] or viscosity [6]. 

Future clinical elastographic diagnosis is thought to be significantly impacted by viscosity [22]. However, so far, published research on viscosity is scarce.

We hypothesize that assessing tissue viscosity might serve as an innovative, non-invasive imaging tool with elastography for the functional MSG assessment. By establishing normal viscosity and elasticity values of MSG, both in a basal state and following stimulation with sialagogue agents, the effect of inflammatory conditions caused by infectious agents, sialolithiasis, or autoimmune disorders, including Sjögren’s Syndrome can be further evaluated. Nevertheless, future research is required to determine the clinical relevance of this new tissue parameter. 

This proof-of-concept study demonstrates that MSG viscosity values presented a positive correlation with the elasticity values in a basal state and following stimulation with citric acid in healthy subjects. However, poststimulation the correlation coefficients were slightly lower for both PG (r = 0.711 vs. r = 0.676) and SMG (r = 0.646 vs r = 0.625). Further studies must assess how these two parameters and their correlation vary in different inflammatory pathologies affecting the MSG. Although viscosity and elasticity are two physically different tissue properties, both finally depend on shear waves (shear wave dispersion influences viscosity; shear wave speed influences elasticity [3]), which explains the partial correlation between these two parameters obtained in our study. Depending on the disease (e.g., inflammation or fibrosis), it is to be expected that these correlations may decrease or no longer exist.

This study has several limitations. Firstly, the measurements were performed using the curvilinear transducer as currently the Vi.PLUS mode is only embedded on this type of transducer. However, valid measurements could be obtained with a Stability Index greater than 90%, which served as the manufacturer’s quality indicator. This study aimed to obtain quantitative information regarding viscosity and elasticity values and not evaluate the structural changes in MSG parenchyma, where high-frequency transducers are compulsory.

Further research is necessary to validate these findings with a linear probe. Secondly, interobserver reproducibility studies must be performed on larger study groups. Individual responses to sialagogue agents must also be considered when performing functional salivary tests [18]. Nevertheless, we assume that further studies are required to verify the level of influence of this new parameter on inflammatory pathologies affecting the MSG.

## 5. Conclusions

In conclusion, Vi.PLUS is a simple, non-invasive, novel technique that allows together with 2D-SWE.PLUS the evaluation of functional changes in major salivary glands in healthy subjects. The generated data might prove helpful in future studies regarding pathological conditions affecting these structures. 

## Figures and Tables

**Figure 1 diagnostics-12-01963-f001:**
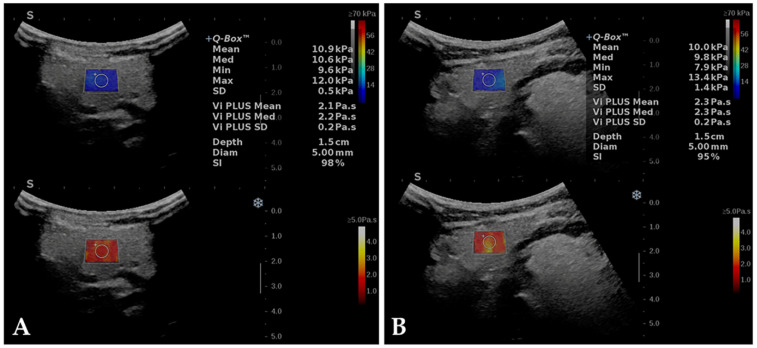
The basal assessment of the parotid gland (**A**) and submandibular gland (**B**) in a healthy subject using 2D-SWE.PLUS (up) and Vi.PLUS (down) modes, simultaneously engaged. A color-coded map is displayed in the 2D-SWE.PLUS box (range 0 to 70 kPa), colors on top of the bar (red) represent high stiffness, while colors on the bottom (blue) represent low stiffness. A color-coded map (range 0 to 5 Pa.s) is also displayed in the Vi.PLUS box, high viscosity is represented by white-yellow colors, while low viscosity is depicted in red.

**Figure 2 diagnostics-12-01963-f002:**
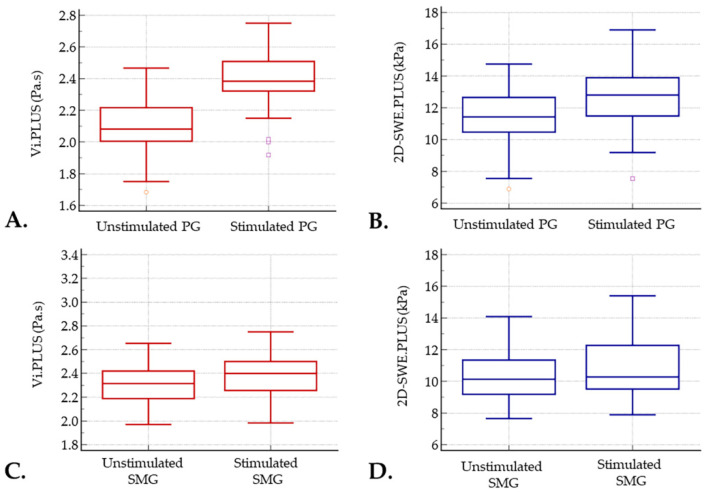
Boxplot showing viscosity (**A**,**C**) and stiffness (**B**,**D**) values of parotid glands (PG) and submandibular glands (SMG) before and after stimulation.

**Figure 3 diagnostics-12-01963-f003:**
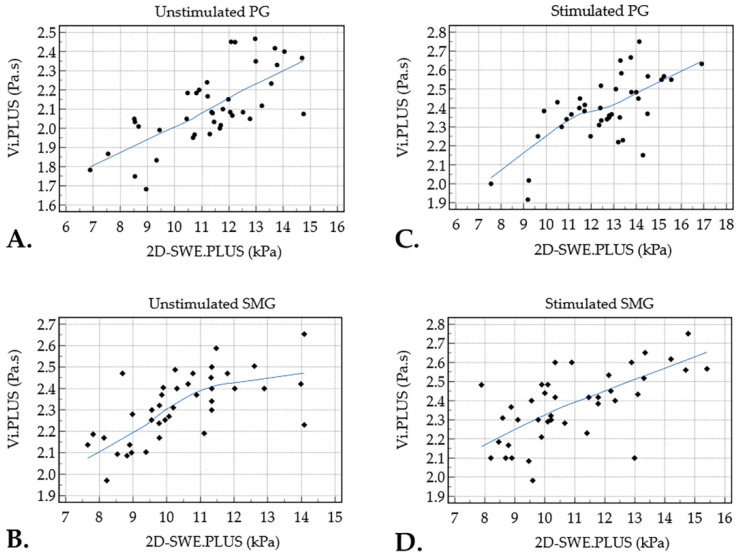
Relationship between the mean values assessed by Vi.PLUS and 2D-SWE.PLUS of parotid glands (PG) and submandibular glands (SMG) before (**A**,**B**) and after stimulation (**C**,**D**) with citric acid in healthy subjects (blue line—trend line).

**Table 1 diagnostics-12-01963-t001:** Descriptive data of the healthy subjects group.

	n (%)/Median [Range]
Total no. subjects	40
Sex	
Female	24 (60)
Male	16 (40)
Age, years	29 [25, 32]
BMI	23 [19.5, 25.8]

n = number of subjects; BMI = body mass index.

**Table 2 diagnostics-12-01963-t002:** Mean stiffness and viscosity values of the parotid and submandibular gland in a group of healthy subjects in a basal state and following stimulation.

		Pre-Stimulation	Post-Stimulation	
Parotid gland				
Viscosity (Pa.s)	Mean	2.10	2.39	***p* < 0.001**
	95% CI	2.04–2.16	2.33–2.44	
	SD	0.19	0.17	
2D-SWE (kPa)	Mean	11.32	12.58	***p* < 0.001**
	95% CI	10.71–11.94	11.96–13.20	
	SD	1.91	1.92	
Submandibular gland				
Viscosity (Pa.s)	Mean	2.31	2.37	*p* = 0.086
	95% CI	2.26–2.36	2.31–2.43	
	SD	0.15	0.18	
2D-SWE (kPa)	Mean	10.40	10.90	*p* = 0.074
	95% CI	9.87–10.93	10.27–11.54	
	SD	1.64	1.98	

CI = confidence interval; SD = standard deviation. Bold values are statistically significant.

**Table 3 diagnostics-12-01963-t003:** Mean viscosity and stiffness values of the parotid and submandibular gland in a group of healthy subjects according to gender, before and after stimulation.

		Pre-Stimulation		*p*	Post-Stimulation		*p*
		Female	Men		Female	Men	
Parotid gland							
Viscosity (Pa.s)	Median	2.07	2.09	0.464	2.38	2.38	0.750
	Q1, Q3	1.98, 2.20	2.02, 2.27		2.33, 2.50	2.30, 2.52	
2D-SWE (kPa)	Median	11.48	11.42	0.955	12.80	12.80	0.955
	Q1, Q3	10.46, 12.65	10.09, 12.59		11.48, 14.05	11.30, 13.36	
Submandibular gland							
Viscosity (Pa.s)	Median	2.30	2.33	0.257	2.39	2.40	0.911
	Q1, Q3	2.15, 2.40	2.26, 2.43		2.17, 2.52	2.3, 2.48	
2D-SWE (kPa)	Median	9.92	10.87	0.163	10.10	11.78	0.103
	Q1, Q3	8.92, 11.11	9.55, 11.74		8.89, 10.78	9.83, 12.61	

Q1 = 25% Quartile; Q3 = 75% Quartile.

**Table 4 diagnostics-12-01963-t004:** Multiple correlations between the body mass index and viscosity and stiffness values of the parotid and submandibular gland, before and after stimulation.

			BMI	*p*
Parotid gland	Viscosity	Pre-stimulation	r = 0.424	0.064
		Post-stimulation	r = 0.121	0.454
	2D-SWE	Pre-stimulation	r = 0.213	0.186
		Post-stimulation	r = -0.163	0.315
Submandibular gland	Viscosity	Pre-stimulation	r = 0.038	0.185
		Post-stimulation	r = 0.242	0.132
	2D-SWE	Pre-stimulation	r = 0.097	0.549
		Post-stimulation	r = 0.116	0.474

BMI = body mass index; r = correlation coefficient.

## Data Availability

The data is available only by request.

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
