# Peer review of "Functional Evaluation of Major Salivary Glands Using Viscosity PLUS and 2D Shear-Wave PLUS Elastography Techniques in Healthy Subjects—A Pilot Study"

_diagnostics, 2022, doi:10.3390/diagnostics12081963_

Round 1
Reviewer 1 Report
Review:
Functional evaluation of major salivary glands using Viscosity 2 PLUS and 2D Shear-Wave PLUS Elastography techniques in 3 healthy subjects – A Pilot Study
Summary:
The authors look at the measurement of viscosity and classical elastography in the parotid and submandibular glands before and after stimulation with lemon juice in healthy volunteers. There were significant differences before and after stimulation for the parotid gland, but not for the submandibular gland.
General:
There was a nice presentation of the study. Nevertheless, there are still a few points that can be improved.
Methodology:
- “Then 5 ml of citric acid as commercially available undiluted lemon juice was given orally using a 5 ml syringe, and 30 seconds later, the measurements were repeated. The lemon juice had to be 78 maintained in the mouth for 10 seconds before being swallowed.” => c Thank you very much for the precise description. Nevertheless, is there more precise data on the lemon juice (pH?, juice concentrate as a base?, added sugar?)?
Results:
- Please check the formatting in table 1. Here the line number 143 is in the table at Age/BMI. Please also check the formatting of table 3. There, the row details are to the left of the table. Possibly a line specification should be omitted from the tables.
- In the subgroup analysis, very small groups are present. Quartiles should therefore be used to describe the group and the results more precisely.
- “Also, no statistically significant correlation was found between the BMI and the vis-199 cosity and elasticity measurements of PG and SMG, in the basal state and following stimulation (p>0.05).” => no results are available here. Please supplement Table 3 accordingly or attach an additional table with the data.
Discussion:
Regarding the correlations: Viscosity and elasticity are physically different. Viscosity describes the thickness in a simplified way. Elasticity, on the other hand, describes simplified how a body returns to its original shape after a force is applied. In this paper, correlations between the two are calculated. However, the question arises as to what the aim of this calculation is. The introduction refers to the possible differences in tissues ("Elasticity is related to shear wave speed, which is linked to tissue fibrosis, while viscosity is related to shear wave dispersion, which proved to be influenced by inflammatory changes"). It is thus almost somewhat reassuring that the correlations are <0.711 and not equal to 1.0. Methodologically, it seems reasonable that there is a correlation because both values now finally depend on the shear wave (elasticity: shear wave speed, viscosity: shear wave dispersion). This can explain the correlation. Depending on the disease (e.g. inflammation or fibrosis), it is to be expected that the correlations may decrease or no longer exist.
Author Response
Thank you very much for the useful observations and constructive feedback.
Point 1:
Methodology: “Then 5 ml of citric acid as commercially available undiluted lemon juice was given orally using a 5 ml syringe, and 30 seconds later, the measurements were repeated. The lemon juice had to be 78 maintained in the mouth for 10 seconds before being swallowed.” => Thank you very much for the precise description. Nevertheless, is there more precise data on the lemon juice (pH?, juice concentrate as a base?, added sugar?)?
Response 1: For the present study we used commercially available 100% lemon juice: undiluted, unsweetened, with no additives or preservatives. No additional information regarding the pH was offered by the producer.
Point 2:
Results: Please check the formatting in table 1. Here the line number 143 is in the table at Age/BMI. Please also check the formatting of table 3. There, the row details are to the left of the table. Possibly a line specification should be omitted from the tables.
Response 2: The formatting of the tables was adjusted.
Point 3:
Results: In the subgroup analysis, very small groups are present. Quartiles should therefore be used to describe the group and the results more precisely.
Response 3: The table regarding the subgroup analysis was changed with a new one presenting the data as median, 25% quartile, and 75% quartile, as suggested.
Point 4:
Results: “Also, no statistically significant correlation was found between the BMI and the vis-199 cosity and elasticity measurements of PG and SMG, in the basal state and following stimulation (p>0.05).” => no results are available here. Please supplement Table 3 accordingly or attach an additional table with the data.
Response 4: An additional table with the data regarding the correlations was attached.
Point 5:
Discussion: Regarding the correlations: Viscosity and elasticity are physically different. Viscosity describes the thickness in a simplified way. Elasticity, on the other hand, describes simplified how a body returns to its original shape after a force is applied. In this paper, correlations between the two are calculated. However, the question arises as to what the aim of this calculation is. The introduction refers to the possible differences in tissues ("Elasticity is related to shear wave speed, which is linked to tissue fibrosis, while viscosity is related to shear wave dispersion, which proved to be influenced by inflammatory changes"). It is thus almost somewhat reassuring that the correlations are <0.711 and not equal to 1.0. Methodologically, it seems reasonable that there is a correlation because both values now finally depend on the shear wave (elasticity: shear wave speed, viscosity: shear wave dispersion). This can explain the correlation. Depending on the disease (e.g. inflammation or fibrosis), it is to be expected that the correlations may decrease or no longer exist.
Response 5: Very useful observation, which was emphasized in the discussion section.
Reviewer 2 Report
Interesting research with possible future application to diagnosis of metabolic diseases.
Author Response
Thank you very much for your constructive feedback.
Reviewer 3 Report
The paper entitled "Functional evaluation of major salivary glands using Viscosity PLUS and 2D Shear-Wave PLUS Elastography techniques in healthy subjects – A Pilot Study" by Muntean and al. evaluates the viscoelastic proprieties of parotid and submandibular glands in a group of 40 healthy subjects using the novel Vi.PLUS and 2D-SWE.PLUS methods. The results show that the mean viscosity and stiffness increased significantly following gustatory stimulation compared to the parotid glands basal state, while for the submandibular glands, the values were only slightly higher, with no statistically significant difference. The manuscript is well written and it is easy to read.
The introduction section deeply describes the study background and the results are clearly presented.
Author Response
Thank you very much for the useful observations and constructive feedback.